# Comparison of Entropy Calculation Methods for Ransomware Encrypted File Identification

**DOI:** 10.3390/e24101503

**Published:** 2022-10-21

**Authors:** Simon R. Davies, Richard Macfarlane, William J. Buchanan

**Affiliations:** Blockpass ID Lab, School of Computing, Edinburgh Napier University, Edinburgh EH10 5DT, UK

**Keywords:** entropy, randomness, crypto-ransomware, mixed file dataset, PRNG

## Abstract

Ransomware is a malicious class of software that utilises encryption to implement an attack on system availability. The target’s data remains encrypted and is held captive by the attacker until a ransom demand is met. A common approach used by many crypto-ransomware detection techniques is to monitor file system activity and attempt to identify encrypted files being written to disk, often using a file’s entropy as an indicator of encryption. However, often in the description of these techniques, little or no discussion is made as to why a particular entropy calculation technique is selected or any justification given as to why one technique is selected over the alternatives. The Shannon method of entropy calculation is the most commonly-used technique when it comes to file encryption identification in crypto-ransomware detection techniques. Overall, correctly encrypted data should be indistinguishable from random data, so apart from the standard mathematical entropy calculations such as Chi-Square ($\chi^{2}$), Shannon Entropy and Serial Correlation, the test suites used to validate the output from pseudo-random number generators would also be suited to perform this analysis. The hypothesis being that there is a fundamental difference between different entropy methods and that the best methods may be used to better detect ransomware encrypted files. The paper compares the accuracy of 53 distinct tests in being able to differentiate between encrypted data and other file types. The testing is broken down into two phases, the first phase is used to identify potential candidate tests, and a second phase where these candidates are thoroughly evaluated. To ensure that the tests were sufficiently robust, the NapierOne dataset is used. This dataset contains thousands of examples of the most commonly used file types, as well as examples of files that have been encrypted by crypto-ransomware. During the second phase of testing, 11 candidate entropy calculation techniques were tested against more than 270,000 individual files—resulting in nearly three million separate calculations. The overall accuracy of each of the individual test’s ability to differentiate between files encrypted using crypto-ransomware and other file types is then evaluated and each test is compared using this metric in an attempt to identify the entropy method most suited for encrypted file identification. An investigation was also undertaken to determine if a hybrid approach, where the results of multiple tests are combined, to discover if an improvement in accuracy could be achieved.

## 1. Introduction

Ransomware infection remains a current and significant threat to both individuals and organisations [1], reinforcing the need for organisations to constantly improve their resilience to such attacks [2]. The research community thus continues to develop robust and effective mitigation techniques. Despite this, recent reports [3] indicate that an increasing number of organisations are succumbing to such attacks and ultimately paying the requested ransom to regain control of their data and affected infrastructures.

Although there are countless strains of ransomware, they mainly fall into two main types. These are crypto-ransomware and locker ransomware. Crypto-ransomware encrypts valuable files on a computer so that they become unusable. Cyber Criminals that leverage crypto-ransomware attacks generate income by restricting access to these files until the victim pays a ransom. Unlike crypto-ransomware, Locker ransomware does not encrypt files. Instead goes one step further, and it locks the victim out of their device. In these types of attacks, cybercriminals will demand a ransom to unlock the device. In both types of attack, users can be left without any option other than payment to recover their files. Over time ransomware attacks have become more sophisticated, performing tasks such as ex-filtrating data and attempting lateral movement to infect other machines. In this paper, we will focus solely on the encryption actions undertaken by crypto-ransomware.

A reoccurring theme within many crypto-ransomware detection techniques is the concept of randomness and file entropy. Researchers assert that a good indicator [4,5,6,7] of crypto-ransomware activity is the generation of files whose contents appears to be random and contain no distinguishable structure. It is agreed that Well-encrypted data should be indistinguishable from random data. A problem with this approach is that the contents of the files created as a consequence of archiving or compression, also tend to have high entropy values, thus interfering with the above assumption. Some modern file formats, such as the new Microsoft Office documents, also employ compression as part of the file format, also increasing their overall entropy. Overall, Shannon entropy appears to be the technique of choice, when researchers wish to determine the randomness of an action or result. An attribute of files encrypted by crypto-ransomware is that while the majority of the contents of the encrypted file contains the encrypted version of the victims file, the file normally also contains extra information relating to the crypto-ransomware strain and how the file was encrypted. This extra information can include check sums, encrypted keys and offset values and generally appears at the start or the end of the file. This extra information could affect the overall entropy profile of the file.

While there exist many techniques available—both purely mathematical, as well as statistical—to determine the entropy or randomness of a file’s contents; in the majority of cases, there is often no discussion in the research into why a specific entropy technique was used over its alternatives. The most common technique used in crypto-ransomware detection systems is to identify files with random content using the Shannon entropy calculation. The closer a file’s overall entropy value is to eight bits, the higher the confidence that its contents are encrypted and subsequently—possibly—the consequence of a crypto-ransomware infection. Other techniques that have been used apart from Shannon entropy are Chi-square, Kullback-Liebler distance, serial byte correlation and Monte Carlo [8].

This paper describes the work performed by the authors in firstly identifying alternative file randomness measurement techniques and then subsequently compares their effectiveness. The work is specifically aimed at measuring the effectiveness of these techniques in being able to correctly distinguish between crypto-ransomware encrypted files and other files that generally have a high entropy value, such as with archived or compressed files. When performing this type of research, achieving convincing and reproducible results [7], is highly dependent on the size, scope and quality of the underlying dataset used. It has been found that seeming successful lab experiments using preselected datasets subsequently perform poorly in real-world evaluations [9]. To address this point, the paper uses the NapierOne [10,11] dataset, due to its broad and comprehensive collection of modern common data types. This dataset also contains large numbers of examples of high entropy files including encrypted, compressed and archived files.

The overall research is broken down into two phases. During the first phase, all of the 53 identified randomness tests were executed against a subset of the NapierOne data set. Techniques that exhibited a reasonably-high accuracy were then selected for inclusion in the second phase. During the second phase of tests, the candidates identified in phase one were re-executed, but against the full NapierOne dataset. This resulted in each randomness test being executed on 5000 examples of each of the 54 data distinct types contained within the dataset. It results in performing nearly three million separate entropy calculations on the NapierOne dataset in an attempt to determine which, if any, of the identified randomness calculations is best suited for crypto-ransomware detection. A subsequent investigation was also performed to determine if a hybrid solution of combining separate entropy calculation results could be used to improve the detection accuracy.

The hypothesis of this paper is that there is a fundamental difference between different entropy methods and that the best methods can be used to better detect ransomware encrypted files.

The contributions of this paper being:The main contribution of this paper is in initially assessing the commonly used entropy measurement methods, and use these to create a short-list of contenders that are assessed on a commonly used data set, for a range of success rate measurements against ransomware/encryption detection.Analyse a large and varied selection of entropy calculation methods in order to determine if any of the techniques are able to accurately distinguish between files generated by a crypto-ransomware and other common high entropy files such as compressed or archive file types.No other published research covers this extensive scope of entropy calculation analysis, using a standard data set, and thus provides a core contribution to crypto-ransomware detection research.

The remainder of the paper is structured as follows. In Section 2, we discuss some of the major uses of statistics in identifying crypto-ransomware infection, as well as previous works which leveraged this approach. In Section 3, we provide a brief description of the tests performed during the research and how the experiments were broken down into two phases. Section 4 explains the methodology used in the experiments and the recorded results. In Section 5, we discuss the consequences of the findings with respect to the development of anti-ransomware techniques, and we provide some recommendations for crypto-ransomware detection approaches moving forwards. Finally, in Section 5, we discuss the main findings and conclusions gained from this research together with possible limitations in using this approach and suggest further research that could be conducted based on the findings from these experiments.

## 2. Related Work

Many crypto-ransomware detection methods use a metric calculated from the file being written to disk as an indicator as to whether the contents of the file are encrypted or not. This calculation is normally used in combination with other identified indicators [6,7] to determine if the file being written to disk is the result of a crypto-ransomware attack. Once an attack has been identified, the system can then decide on an appropriate response, such as retaining copies of the encryption keys [12,13], informing the users and/or terminating the processes [14,15] that initiated the write. This calculation may be performed directly on the file being written or alternatively the difference between the calculated value of the file being read and subsequently written.

Since its discovery in 1948 [16], Shannon entropy has been successfully applied in many fields of research. The majority of its application is in the determination of information, or conversely uncertainty inherent in a variable. Within the field of malware research and more specifically crypto-ransomware research, this technique has been applied in various contexts. For example, in the measurement of randomness in the behaviour of the crypto-ransomware code such as in the use of system calls (API’s) [17] or in the content of files generated by these programs. Files with a simple format in their structure and contents, such as plain text files, tend to have higher predictability and hence lower entropy than files that have been compressed or encrypted. This difference in entropy can be used as an indicator to identify when encrypted files, or more specifically high entropy files, are being written to disk. Significant research has been performed into crypto-ransomware detection using the Shannon entropy calculation [6,13,14,15,17,18,19,20,21,22,23,24,25,26,27,28,29], resulting in many interesting detection techniques and tools. However, some researchers do comment on the unsuitability of this technique when analysing typically higher entropy files [7,30] such as with archive and compressed files.

While Shannon’s entropy has been the technique of choice in the majority of reviewed research, some researchers have used alternative techniques to identify the randomness of a file’s contents. These alternative techniques include: chi-square calculation [18,31,32,33,34,35]; the Kullback-Liebler [30] technique; and Serial byte correlation [8,36]. In the majority of research though, no specific reason is often provided as to why the specific entropy calculation is selected and also it is common that the evaluation was performed on an undefined dataset of limited scope and variety of file types.

Apart from these *pure* mathematical techniques where the randomness of the data is determined from a single calculation, there also exist alternative approaches, such as in the test suites that have been developed to validate pseudo-random number generation (PRNG) programs. The aim of random number generators is to produce a series of numbers that do not display any distinguishable patterns in their appearance or generation. The output of these random number generators should thus have very high entropy, which would then lend the suites used to test the outputs of these generators suitable to also test the contents of encrypted files. There exist many test suites currently available to test the output of random number generators, the most significant being: NIST SP 800-22 [37], the Chinese equivalent to the NIST test GM/T 0005-2012 [38,39], FIPS-2-140 [37,40], DieHarder [41,42] and the TESTU01 [43,44] libraries.

## 3. Methodology

The main focus of this paper is to compare multiple entropy evaluation techniques using a large and varied dataset, in an attempt to determine which tests, if any, exhibit the highest accuracy in differentiating between crypto-ransomware encrypted files from other common file types. The NapierOne data set [10,11] was selected as it fulfils the requirement that the test dataset contains a large selection of commonly used file types including multiple examples of compressed, archive and encrypted formats. During the provisional research, it was identified that some of the randomness tests took a significant amount of time to complete, so it was decided to divide this research into two phases. Firstly, in the qualification phase - all the identified tests were executed on a smaller subset of files, taken from the NapierOne dataset. This research is aimed at using randomness identification techniques to differentiate between crypto-ransomware encrypted files and other file types. The specific file types used to populate the Phase 1 data subset were selected according to the formats that typically are well-known for having higher entropy [7,14]. Tests that produced a true positive rate of above 80% and were completed within a reasonable amount of time, and were then selected to participate in the more extensive second phase of testing. The reasoning for this, was that there was little point in testing further, techniques that are either slow to execute or provide unreliable results.

### 3.1. Proposed Tests

The following tests and test suites were considered as part of the first phase of testing. Tests that performed well with regards to accuracy and performance were later executed in Phase 2 of the tests.

#### 3.1.1. NIST SP-800 22

The NIST SP800-22 specification [37,40,45] from 2010 describes a suite of tests whose intended use is to evaluate the quality of random number generators [21]. The suite consists of 15 distinct tests, and which analyse various structural aspects of a byte sequence. These tests are commonly employed as a benchmark for distinguishing compressed and encrypted content (e.g., [18,31]). Each test analyses a particular property of the sequence, and subsequently applies a test-specific decision rule to determine whether the result of the analysis suggests randomness, or not.

Since the tests only consider the output values and not the implemented methods for generation, some researchers argue that the tests contained within this suite are not useful [27], arguing that the evaluation of pseudo-random generators and sequences should be based on cryptanalytic principles. Implementation validation should be focused on algorithmic correctness, not the randomness of the output. However, it was decided to still incorporate these tests into our evaluation due to their universal acceptance and use within the field of randomness testing. In January 2022, NIST confirmed [46] that the test suite will be reviewed and possibly updated in the future.

The specific NIST tests used during the first phase of testing are: 

**Frequency (Monobit) Test.** Determine whether the number of ones and zeros in an entire sequence is approximately the same as would be expected for a truly random sequence.**Frequency Test within a Block.** Determine whether the frequency of ones in an *M*-bit block is approximately $\frac{M}{2}$, as would be expected under an assumption of randomness.**Runs Test.** Test the total number of runs in the sequence, where a run is an uninterrupted sequence of identical bits. The purpose of the Runs test is to determine whether the number of runs of ones and zeros of various lengths is as-expected for a random sequence. In particular, this test determines whether the oscillation between such zeros and ones is too fast or too slow.**Test for the Longest Run of Ones in a Block.** Determine whether the length of the longest run of ones within the tested sequence is consistent with the length of the longest run of ones that would be expected in a random sequence.**Binary Matrix Rank Test.** Check for linear dependence among fixed-length sub-strings of the original sequence. Note that this test also appears in the DIEHARD suite of tests [47,48].**Discrete Fourier Transform (Spectral) Test.** Detect periodic features (i.e., repetitive patterns that are near each other) in the tested sequence that would indicate a deviation from the assumption of randomness. The intention is to detect whether the number of peaks exceeding the 95% threshold is significantly different than 5%.**Non-overlapping Template Matching Test.** Count the number of occurrences of pre-specified target strings and identifying too many occurrences of a given non-periodic (aperiodic) pattern. An example of an 68-bit aperiodic pattern being 0 1 1 1 1 1 1 1.**Overlapping Template Matching Test.** Similar to the previous test, this test also looks for occurrences of pre-specified target strings. When a match is found, this test moves the test window by one byte, whereas the previous test moves the test widow to the end of the matching sequence.**Maurer’s *Universal Statistical* Test.** This is the number of bits between matching patterns (a measure that is related to the length of a compressed sequence). The purpose of the test is to detect whether or not the sequence can be significantly compressed without loss of information. A significantly compressible sequence is considered to be non-random.**Linear Complexity Test.** This tests the length of a linear feedback shift register (LFSR). The linear complexity of a sequence equals to the length of the smallest linear feedback shift register (LFSR) that generates the given sequence. Random sequences are characterised by longer LFSR’s. An LFSR that is too short implies non-randomness.**Serial Test.** Test the frequency of all possible overlapping *m*-bit patterns across the entire sequence.**Approximate Entropy Test.** Similar to the previous test, this tests the frequency of all possible overlapping m-bit patterns across the entire sequence. The purpose of the test is to compare the frequency of overlapping blocks of two consecutive/adjacent lengths (*m* and *m* + 1) against the expected result for a random sequence.**Cumulative Sums (Cusum) Test.** The focus of this test is the maximal excursion (from zero) of the random walk defined by the cumulative sum of adjusted (−1, +1) digits in the sequence. For a random sequence, the excursions of the random walk should be near zero.**Random Excursions Test.** Test the number of cycles having exactly *K* visits in a cumulative sum random walk. This test is to determine if the number of visits to a particular state within a cycle deviates from what one would expect for a random sequence.**Random Excursions Variant Test.** Test the total number of times that a particular state is visited in a cumulative sum random walk. The purpose of this test is to detect deviations from the expected number of visits to various states in the random walk.

#### 3.1.2. Dieharder2

This test suite from 2006 is a public licensed implementation [41,42] of the tests developed by Marsagla [47] as part of the original Diehard test suit [49]. To aid readability of the paper the descriptions of the *Dieharder2* tests have been placed in the Appendix A.

#### 3.1.3. PractRand

Is a C++ library for convenient random number generation based on a variety of high-quality algorithms; and also a C++ library of statistical tests for RNGs [43,50,51,52]. It includes a standard battery of tests, in the tradition of Diehard, that can detect bias in a wide variety of RNGs quickly.

#### 3.1.4. Mathematical Tests

**Shannon Entropy** File entropy refers to a specific measure of randomness. One such measure is called *Shannon Entropy* [16] and is used to express information content. This value is essentially a measure of the predictability of any specific byte in the file, based on preceding bytes [53]. It is basically a measure of the *randomness* of the data in a file—measured in a scale of zero to eight (eight bits in a byte). Typical text files that contain only alphanumeric characters and no formatting will have a lower value, whereas encrypted or compressed files will have a value closer to 8 [54].

Another way to consider entropy is that it is a measure of the predictability or randomness of data. A file with a highly predictable structure or a bit pattern that repeats frequently has low entropy. Such files would be considered to have low information content (or low information density). Files in which the next byte value is relatively independent of the previous byte value would be considered to have high entropy. Such files would be thought to have high information content [55,56].

The maximum possible entropy per byte is eight bits of entropy per byte signifying that it is completely random. The generally accepted formula for entropy (*H*) [16], is shown in Equation (Equation 1) and were calculated using the ent [57] program:(1)$$H\left(X\right)=-\sum_{i=1}^{n}P\left(x_{i}\right)\;log_{2}\;P\left(x_{i}\right)$$where

*H* is the entropy (measured in bits)*n* number of bytes in the sample$P(x_{i})$ probability of byte *i* appearing in the stream of bytes.

The negative sign ensures that the result is always positive or zero.$X^{2}$**Chi-square.** The chi-square (*X*${}^{2}$) test is a popular statistical test generally used to determine if an observed distribution is statistically similar to an expected distribution [58]. In the case of perfectly random data on the file system, we would expect an equal occurrence of each byte value. In the case of crypto-ransomware detection, where the files are encrypted, we use equal frequencies of byte values for the expected distribution, and use the formula presented in Equation (Equation 2), to check if the observed distribution is similar [57,59]:(2)$$x^{2}=\sum_{i=0}^{255}\frac{{\left(O_{i}-E_{i}\right)}^{2}}{E_{i}}$$
where$x^{2}$ = chi squared$O_{i}$ = Observed value$E_{i}$ = Expected value

**Monte Carlo.** This is a technique of approximating the value of a function where calculating the actual value is difficult or impossible. It uses random sampling to define constraints on the value and then makes a sort of *best guess* [60]. The underlying concept is to use randomness to solve problems that might be deterministic in principle. Monte Carlo values are calculated using the formula presented in Equation (Equation 3) and were calculated using the ent [57] program.
(3)$$E\left(X\right)\approx \frac{1}{N}\sum_{n=1}^{N}x_{n}$$
where*E* = Result of approximation$x_{n}$ = Randomly chosen value

**Arithmetic Mean.** The mean is the average or the most common value in a collection of numbers and is calculated by adding all the numbers in a given data set and then dividing it by the total number of members of that data set. The formula used to calculate this value is presented in Equation (Equation 4) and were calculated using the ent [57] program.
(4)$$M=\frac{S}{T}$$
whereM = Arithmetic MeanS = Sum of ObservationsT = Sum of Observations

**Serial Byte Correlation Coefficient.** A lightweight statistical test that looks at the relationship between consecutive numbers. This quantity measures the extent to which each byte in the file depends upon the previous byte. Tests the correlation between bytes written to a file, expecting a low value for encrypted files [8]. For random sequences, this value (which can be positive or negative) will, of course, be close to zero. The Serial Correlation Coefficient of a sequence is calculated using the formula given by Knuth [61]. For a sequence of numbers $U_{0}$, $U_{1}$, $U_{2}$,....., $U_{n-1}$, the Serial Byte Correlation Coefficient is defined as shown in Equation (Equation 5) and was calculated using the ent [57] program:(5)$$C=\frac{n\left(U_{0}U_{1}+U_{1}U_{2}+\cdots +U_{n-2}U_{n-1}+U_{n-1}U_{0}\right)-{\left(U_{0}+U_{1}+\cdots +U_{n-1}\right)}^{2}}{n\left(U_{0}^{2}+U_{1}^{2}+\cdots +U_{n-1}^{2}\right)-{\left(U_{0}+U_{1}+\cdots +U_{n-1}\right)}^{2}}$$
**Kullback-Liebler(KL).** KL divergence, which is closely related to relative entropy, information divergence, and information for discrimination, is a non-symmetric measure of the difference between two probability distributions and has been suggested as a good test [30], especially when considering high entropy files that are not encrypted. The equation for this calculation is shown in Equation (Equation 6).
(6)$$D\left(px\|qx\right)=\sum_{i=1}^{n}log\left(\frac{px(x_{i})}{qx(x_{i})}\right)\times px\left(x_{i}\right)$$
where
D(px‖qx) is the relative entropy*n* is the number of bytes in the samplep(x) first probability distributionq(x) second probability distribution

#### 3.1.5. Other Tests Considered

**TESTU01** A popular [43] software library implemented in the ANSI C language and offers a collection of utilities for the empirical statistical testing of uniform random number generators (RNGs) including several pre-configured groups of tests called SmallCrush, Crush and BigCrush [44]. A significant amount of effort was spent attempting to utilise this library to perform statistical analysis on bitstreams, unfortunately without success as the sourced package for this library would not compile on the authors’ platform.**FIPS-2-140** [37,40] Is reported to be a more robust methodology than SP 800-22 test suite [18] but no actual implementation of the tests could be sourced [27].**GM/T 0005-2012** [38,39] The current Chinese standard GM/T 0005-2012 contains a very similar set of tests to the NIST SP-800 22 tests and it is expected that the newer version GM/T 0005-2021 will also be the same. Actual coded examples of these tests were unable to be sourced [27].

### 3.2. Experimental Overview

The main focus of this paper is aimed at testing techniques that can successfully differentiate between encrypted files and other common high entropy files. For each test performed, a specific threshold will be defined. If the outcome of the test performed is within the defined threshold then the tested file will be classified as encrypted, otherwise, if the results fall outside the threshold, the file will then be classified as unencrypted. No further classification attempts will be made. A basic outline of the experimental steps performed is illustrated in Figure 1.

### 3.3. Classification

When classifying the file types, there are four possible outcomes, which are presented in Table 1.

The proposed model uses these classification values to determine the overall accuracy of the test [62]. The formula to compute the Accuracy (ACC) is shown in Equation (Equation 7).
(7)$$\mathrm{ACC}=\frac{TP+TN}{TP+TN+FP+FN}$$ The main success of the test will be judged using the calculated accuracy value, however the following values will also be calculated for completeness.

**Recall** Shown in Equation (Equation 8), is used to calculate how often the test correctly predicted the result.
(8)$$\mathrm{REC}=\frac{TP}{TP+FN}$$
**Precision** Shown in Equation (Equation 9), is used to calculate how often the test is correct.
(9)$$P=\frac{TP}{TP+FP}$$
**F1** Shown in Equation (Equation 10), is used to calculate how balanced the test is between the precision and the recall values.
(10)$$F1=2*\frac{Precision\times Recall}{Precision+Recall}$$

## 4. Implementation and Evaluation

To perform a comprehensive realistic reliable test, the NapierOne data set was used [10,11]. Using this dataset allowed each of the identified tests to be performed on large collections of many of the most common modern data types currently in use today. The data set contains 5000 examples of each of the data types used during testing.

Unlike the research performed by Casino [18] where a small subset of SP 800-22 tests was selected for analysis, in this research all the tests present in the NIST, Dieharder2 and PracRand test suites as well as the mathematical tests described in Section 3.1.4 were evaluated.

As more than 50 possible randomness tests have been identified as candidate techniques for differentiating between encrypted and compressed files, it was decided to divide the comparison testing into two phases. In the first phase, all the identified tests will be run against a subset of the data set. During this phase, tests that exhibit promising results were identified and were selected to be re-executed in the second phase of testing.

### 4.1. Phase one Testing, Qualification

In the initial testing phase, all the candidate tests were executed against a subset of the main NapierOne dataset. File types that tend to have relatively high entropy values [7,55] such as compressed archives, or document formats that utilise compression (e.g., DOCX), were specifically selected to be included in this phase one data subset. The rationale is, that as this research is intended to identify techniques that were able to differentiate between high entropy files and encrypted files, it would be advantageous to identify potentially accurate tests early. However, some specific tests were permitted to proceed to the second phase despite their poor performance. The intent being, that due to these test being widely used within crypto-ransomware detection systems, the progression of these test would results in findings that would be more beneficial to the research community. 20 files of each file type were selected from the NapierOne data set to create the dataset used during the phase 1 testing. An overview of the dataset used, together with the size of each subset, is shown in Table 2.

While it is normally quite straightforward to use entropy to differentiate between encrypted files and other more common lower entropy file types when analysing the Shannon entropy values for the files contained in the phase one dataset, it can be seen that this approach could be quite problematic as the files in this dataset all have high entropy values (the maximum being 8). Just relying on the calculated Shannon entropy value to distinguish between encrypted and non-encrypted files would thus be a difficult task generating a lot of false positives and negatives. The entropy distribution for the file types in this dataset is shown in Figure 2. When examining this figure it can be seen that the average Shannon entropy of this dataset is, as designed, relatively high.

The resulting value calculated from the execution of the test must be within the threshold, for the file to be classified as encrypted and outside the threshold for the file to be classified as unencrypted. The thresholds used for each of the tests are shown in Table 3. For a test to be considered a candidate for qualification to the second phase of testing, it was decided that it needed to meet two criteria. Firstly it must have a classification accuracy of 80% or above when deciding if the file is encrypted or not. The test must also achieve this accuracy for 85% of the entire data set. If the test has a lower accuracy than this, then it is considered not a viable detection technique. The reasoning for this is that there is little purpose in testing further any techniques that already exhibit a low accuracy. Secondly, it must have sufficient performance to allow it to be used in real-world situations. Even if the technique under investigation, is extremely accurate, if it takes a significant amount of time to produce a result, it would prove prohibitive for it to be used and real-time detection systems. The throughput speed of the test will be measured and recorded as processed megabytes per second, or throughput (MB/s).

### 4.2. Results from Phase One Testing

During the first round of testing, 53 distinct tests were performed against a dataset of 320 files, providing nearly 17,000 results. The accuracy of each test performed was then calculated and recorded. The results for each test against each data type are shown in Figure 3. Tests that were 100% accurate in differentiating between crypto-ransomware encrypted and non-encrypted files are represented in the diagram as a green block, if the test had an accuracy of between 80% and 100%, they are represented as an orange block, and tests that had an accuracy below 80% are represented as a red block. Tests that contained many instances of low accuracy were subsequently relegated and not considered in the phase 2 tests.

The performance of the tests was also recorded and these results are presented in Figure 4. The columns of Figure 3 and Figure 4 are aligned so it is possible to view these diagrams together to gain a better overall understanding of the results.

The accuracy of the tests performed was then reviewed and the overall time taken for the completion of the test was also taken into consideration. Using these criteria, 42 of the original tests were rejected, resulting in 11 tests being selected for deeper examination in Phase 2 of the testing. Despite some of the tests exhibiting poor accuracy results, it was decided to also include several of the purely mathematical calculation tests, in the second phase of testing anyway. This was due to the fact that they have been used, or mentioned frequently, in other research and the authors thought that a robust evaluation of these calculations would provide a useful reference. A full list of the tests performed and if they qualified for the second phase of testing is presented in Table 4.

### 4.3. Phase Two Testing

The tests identified from the first phase of testing will then be run against 5000 example files for each of the file types present in the NapierOne data set. The file type contained within this data set is shown in Table 5.

As well as running the selected tests against the standard common file types present in the NapierOne dataset, the tests will also be executed against 5000 example files that have been encrypted by each of the following 12 crypto-ransomware strains: BlackMatter, Conti, DarkSide, Dharma, GandCrab, HelloKitty, Maze, Netwalker, NotPetya, Phobos, Ryuk and Sodinokibi (see Table A1. Most modern crypto-ransomware strains implement hybrid crypto systems [63] where a combination of symmetric and asymmetric encryption is used. In the most modern ransowmare strains a separate symmetric key is generated for each file that is encrypted. The reason for this is that symmetric encryption is significantly quicker than asymmetric encryption, thus allowing the crypto-ransomware to complete its task faster, ideally before the crypto-ransomware is detected. Once the file has been encrypted, the symmetric key used in the encryption is then itself encrypted using the attacker’s public, asymmetric key. There is some variation between the different crypto-ransomware strains relating to the symmetric encryption algorithm used, and due to this, the entropy experiments attempted to use a sample of the most common victim file encryption techniques.

While both Dharma, HelloKitty and Ryuk use AES, Dharma uses CFB mode [63,64] and Ryuk uses CBC mode [65,66]. The AES mode used by HelloKitty is unknown [67]. NetWalker [63,68], NotPetya [69], GandCrab [70] and Sodinokibi [63] use a standard Salsa20 encryption algorithm and BlackMatter and Darkside use a custom Salsa20 matrix [71]. Finally the Maze crypto-ransomware uses ChaCha [63,72] and Conti switched from originally using AES to now also use ChaCha [73]. Care was taken in the selection of the crypto-ransomware stains selected for testing, so that there were several examples present in the dataset of all the encryption algorithms most often used by crypto-ransomware.

### 4.4. Results from Phase Two Testing

During this phase the entire NapierOne dataset was used, resulting in 5000 example files for each of the 42 main data types being tested (Table 5). Additionally, included in the test data set were examples of files that had been encrypted by 12 seperate crypto-ransomware strains (BalckMatter, Conti, DarkSide, Dharma, GandCrab, HelloKitty, Maze, Netwalker, NotPetya, Phobos, Ryuk and Sodinokibi). 5000 encrypted example files for each of these crypto-ransomware strains were also included in the test dataset. This resulted in a total test dataset of 270,000 distinct files. Each of the 11 tests selected from phase one was then re-executed against this enlarged dataset, resulting in a total of nearly three million calculations.

The results of these calculations were then recorded and the classification accuracy for each test, broken down by file type, was calculated. The test’s precision, recall and F1 values were calculated on a per-test basis and the results are shown in Table 6. While the accuracy calculation results when performed on file types with a lower entropy are presented in Table 7. The results for the calculations on file types with higher entropy are presented in Table 8. To aid readability the highest accuracy results have been highlighted in green.

For the lower entropy file type, the purely mathematical calculations appear to perform better than the test suite tests, while for the higher entropy file types it can be seen that no technique was ideally suited to differentiating between crypto-ransomware encrypted files and compressed files, it was noticed that some tests performed better than others. For this testing it can be seen that the most consistently accurate tests were: block frequency, sums and SP-800 Serial correlation from the test suite and Chi-square as a purely mathematical calculation. These findings tend to agree with [8] that Shannon entropy is not a particularly good indicator of encryption when testing archived files, rather a more robust mathematical technique appears to be Chi-Square.

It can also be noted that the Serial Byte Correlation Coefficient tests had an extremely high accuracy in identifying compressed files with almost 100% TN rate on all the compressed file formats. After observing this characteristic some further testing was performed. An evaluation was made to determine if the use of a combination of a purely mathematical calculation together with the results of the Serial Byte Correlation Coefficient calculation, could be used to improve the overall classification accuracy. Using these two techniques in a collaborative approach. In essence, the Serial Byte Correlation Coefficient calculation would be used to determine if the file was an archive and thus improve the TN rate and reduce the FP rate of the purely mathematical calculation.

Firstly, the primary entropy value was calculated, using one of the pure mathematical techniques such as Shannon, Chi-Square or Monte Carlo. Then the Serial Byte Correlation Coefficient was calculated on the same file to determine if the file was an archive. If the Serial Byte Correlation Coefficient test confirmed that the file was actually an archive then any decision made by the first calculation, that the file is encrypted, is reversed and the conclusion of the combined test is that the file is not encrypted. However, if the Serial Byte Correlation Coefficient calculation confirms that the file is not an archive then any decision made by the pure mathematical test, that the file is encrypted, is upheld. This should have the effect of reducing the FP results for the primary calculation on archive files, thus raising the overall accuracy. Table 9 shows the results of this calculation combination test, while it can be seen that the accuracy has been improved over the generic calculations, they are still well below the figures generated by some of the selected SP-800 tests.

Figure 5 demonstrates that it is unfortunately not feasible to use just the serial correlation calculation results in isolation, as this technique did not work well at identifying crypto-ransomware encrypted files. The blue dots in the scatter plot represent all the files in the entire dataset, while the red dots represent the encrypted files. It can be seen that while there is a tendency for encrypted files to have a serial correlation close to zero, it is obvious that this is not always the case.

The final technique evaluated was to combine the five pure mathematical calculations: Shannon, Chi-Square, Mean, Monte Carlo and Serial Byte Correlation Coefficient. The values for each of these techniques was calculated and then a majority voting approach was adopted. If three or more of these calculations concluded that the file was encrypted, then this was the result given, otherwise, the file is deemed to be unencrypted. Unfortunately, this approach had very little impact on detection accuracy.

## 5. Conclusions

From the research performed and the results achieved, it is clear that the use of pure mathematical techniques such as Shannon entropy, Chi-square and Monte Carlo calculations as an indicator for crypto-ransomware encrypted files is not ideal. Especially when applying these calculations to a broad and modern data set, such as NapierOne, it can be seen that these calculations struggle to differentiate consistently between encrypted files and other high entropy files such as compressed or archived files. However, of the purely mathematical calculations tested, then the results from the Chi-Square calculation appear to produce the highest accuracy.

For certain file types, for example, the files created from the execution of the NetWalker crypto-ransomware, all the techniques tested, were unable to successfully identify even 50% of the files as being encrypted. This is interesting as this crypto-ransomware uses the Salsa20 symmetric encryption algorithm, which is the same encryption algorithm used by both the NotPetya and Sodinokibi crypto-ransomware strains.

In certain circumstances, a higher detection accuracy of crypto-ransomware encrypted files was achieved, by using a combination of pure mathematical calculations together with the results produced by the serial byte correlation coefficient calculation. Another interesting discovery was that no tests from the DieHarder2 test suite were accurate enough for the identification of crypto-ransomware encrypted files and no test from this suite qualified the phase 2 test round. Even though the documentation for the Dieharder2 test suite state that some of the tests were similar to the test found in the SP 800-22 test suite, the results did not reflect this.

To the best of the authors’ knowledge, a comprehensive test incorporating both pure mathematical calculations as well as statistical test suite analyses when applied to crypto-ransomware encryption identification has not been performed, and it is hoped that the results from this investigation would prove useful to crypto-ransomware researchers when designing their detection techniques and systems.

### 5.1. Limitations

Using an entropy value as an indication of crypto-ransomware infection has been a common feature in ransomware detection systems for many years, however, some previous research [7], and more recently [74], have investigated techniques that could be used by crypto-ransomware developers to avoid creating files with an elevated entropy value and thus avoid detection using these techniques. One suggested approach is to further encode the encrypted files in a way as to reduce their overall entropy value, by for example, using base64 encoding. However, despite this theoretical mitigation technique, crypto-ransomware techniques are still being proposed that utilise entropy calculations as part of their design [21,75,76,77,78].

The authors of these entropy reduction technique [7,74] state in their papers that they are currently theoretical techniques that could be used by ransomware, but to date, the authors have not encountered any crypto-ransomware strain that have employed them, suggesting that the entropy calculation detection avoidance techniques remain theoretical. This does not mean that these techniques will not be deployed in the future so one area of research would be to update the detection techniques so that they can identify these entropy reducing methods and then adapt their entropy calculation to take this in to account.

### 5.2. Future Work

There exist techniques that can be used to reduce the overall entropy of a file generated by crypto-ransomware [7,74] so one area of research would be to investigate methods to detect if these techniques have been employed, and if so, then modify the entropy calculation to take this into account.

Some interesting outcomes from this research which could be investigated further are why all the techniques struggled in identifying files generated by the NetWalker crypto-ransomware despite it using the same symmetric encryption algorithm as other tested crypto-ransomware. Furthermore, why were the tests from the DieHarder2 test suite not effective in performing this crypto-ransomware encryption classification? Finally, further investigation into the definition of the threshold for Chi-Square calculation may be performed to determine if modification of this could improve the overall accuracy of the crypto-ransomware encrypted file identification calculation.

## Figures and Tables

**Figure 1 entropy-24-01503-f001:**
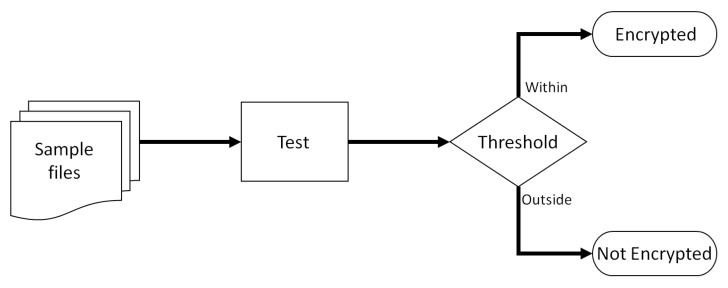
Experiment overview.

**Figure 2 entropy-24-01503-f002:**
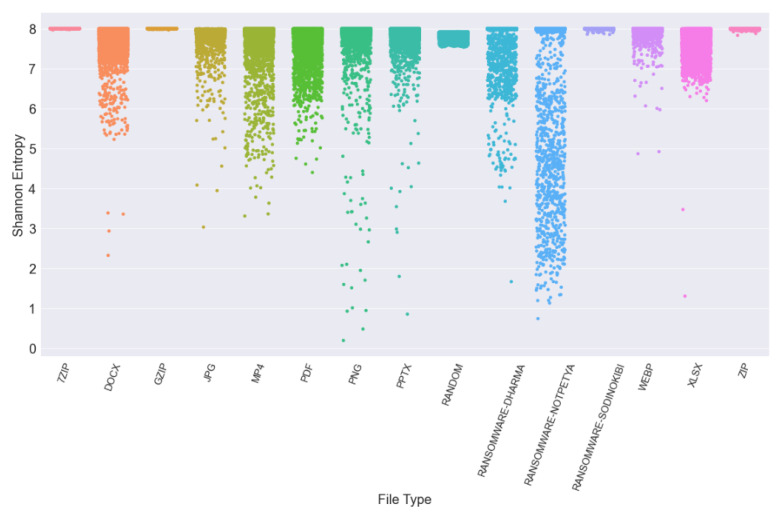
Shannon entropy plot.

**Figure 3 entropy-24-01503-f003:**
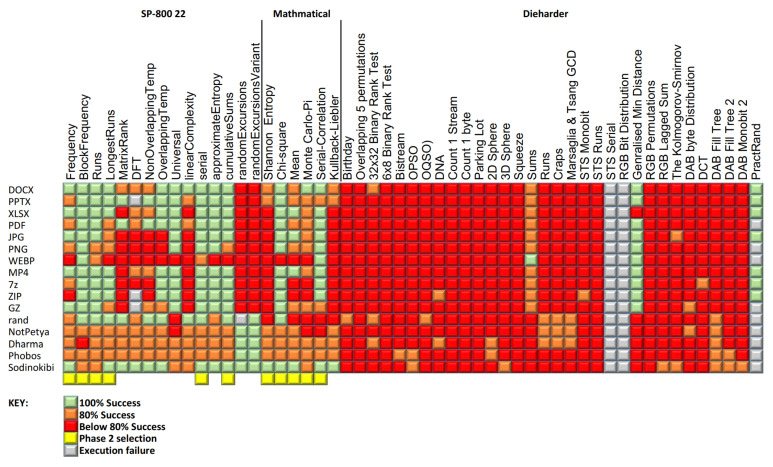
Phase 1 accuracy results.

**Figure 4 entropy-24-01503-f004:**
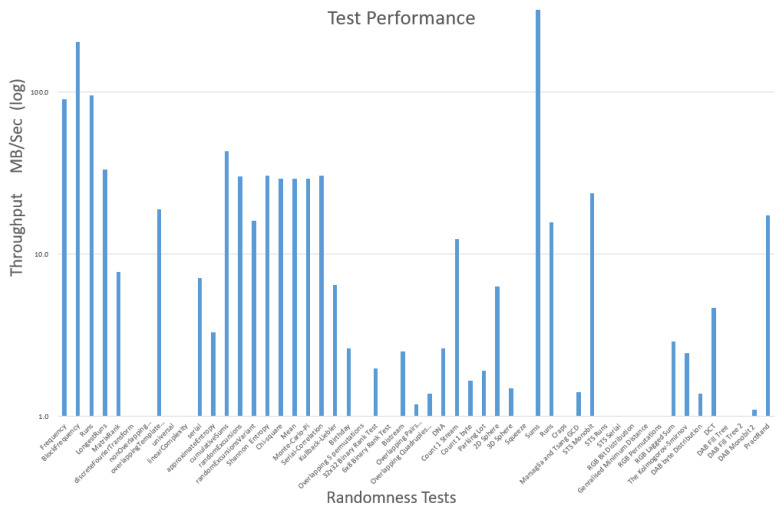
Phase 1 timings.

**Figure 5 entropy-24-01503-f005:**
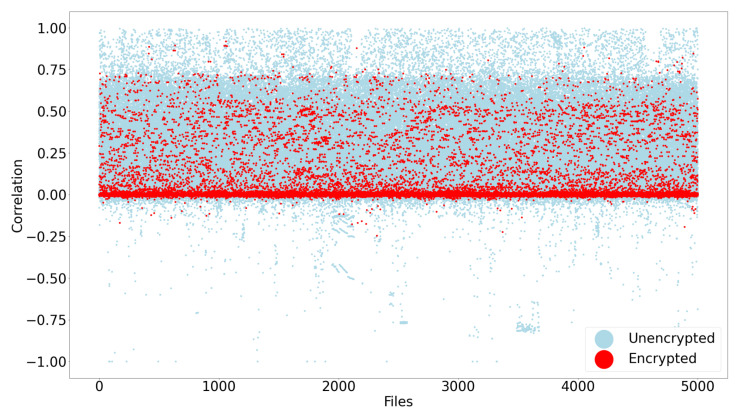
Scatter plot for serial byte correlation coefficient results.

**Table 1 entropy-24-01503-t001:** Possible classification outcomes.

Classification	Description
True Positive (TP)	Correctly classified encrypted file
True Negative (TN)	Correctly classified non-encrypted file
False Positive (FP)	Classified normal file as an encrypted file
False Negative (FN)	Failed to classified encrypted file

**Table 2 entropy-24-01503-t002:** Provisional Test Data Set.

Type	Dataset Size (MB)	Description
DOCX	3.9	Document files
PPTX	73.4	Document files
XLSX	16.6	Document files
PDF	24.3	Document files
JPG	1085.0	Images files
PNG	2.0	Images files
WEBP	2.4	Images files
MP4	30.6	Media files
7ZIP	131.6	7zip compressed files
ZIP	131.4	WinZip compressed files
GZ	131.6	gzip compressed files
Dharma	28.1	Ransomware encrypted files
NotPetya	26.9	Ransomware encrypted files
Phobos	28.1	Ransomware encrypted files
Sodinokibi	25.9	Ransomware encrypted files

**Table 3 entropy-24-01503-t003:** Individual Test Thresholds.

Test	Threshold	Description
SP-800 22	0.01	Probability greater than [45]
Dieharder2	0.5	Probability greater than [41]
Shannon Entropy	7.95	Bit value greater than [30]
Chi-Square	0.01	Probability greater than [57]
Monte Carlo	0.015	Within range +/− π [57]
Mean	0.85	+/− 127.5 [57]
Serial Correlation	0.0011	Correlation less than [57]
Kullback-Liebler	0.01	Greater than [30]
PractRand	0.01	Greater than [50]

**Table 4 entropy-24-01503-t004:** Test selection.

Test	Qualified	Comments
Frequency	Yes	
BlockFrequency	Yes	Tests qualified due to their & performance and accuracy.
Runs	Yes	
LongestRuns	Yes	
SP-800 Serial	Yes	
CumulativeSums	Yes	
Chi-square	Yes	
Serial ByteCorrelation	Yes	
Shannon Entropy	Yes	Accepted due to their overall use
Mean	Yes	
Monte Carlo-Pi	Yes	
ApproximateEntropy	No	
MatrixRank	No	
DiscreteFourierTransform	No	
NonOverlappingTemplate	No	
OverlappingTemplate	No	
Universal	No	
LinearComplexity	No	
RandomExcursions	No	
RandomExcursionsVariant	No	
Kullback-Liebler	No	
Birthday	No	Rejected due to low performance and/or lack of accuracy
Overlapping 5 permutations	No	
32 × 32 Binary Rank Test	No	
6 × 8 Binary Rank Test	No	
Bistream	No	
OPSO	No	
OQSO	No	
DNA	No	
Count 1 Stream	No	
Count 1 byte	No	
Parking Lot	No	
2D Sphere	No	
3D Sphere	No	
Squeeze	No	
Sums	No	
Runs	No	
Craps	No	
Marsaglia and Tsang GCD	No	
STS Monobit	No	
STS Runs	No	
STS Serial	No	
RGB Bit Distribution	No	
Genralised Minimum Distance	No	
RGB Permutations	No	
RGB Lagged Sum	No	
The Kolmogorov–Smirnov	No	
DAB byte Distribution	No	
DCT	No	
DAB Fill Tree	No	
DAB Fill Tree 2	No	
DAB Monobit 2	No	
PractRand	No	

**Table 5 entropy-24-01503-t005:** NapierOne file types.

Type	Type	Type	Type
7ZIP	EPUB	MP3	SVG
APK	EXE	MP4	TIF
BIN	GIF	ODS	TXT
BMP	GIF	OXPS	WEBP
CSS	GZIP	PDF	XLS
CSV	HTML	PNG	XLSX
DOC	ICS	PS	XML
DOCX	JS	PPT	ZIP
DWG	JPG	PPTX	ZLIB
ELF	JSON	RAND	
EPS	MKV	RAR	

**Table 6 entropy-24-01503-t006:** Individual Test Performance Metrics.

	Accuracy	Recall	Precision	F1
BlockFrequency	0.90	0.71	0.86	0.78
Frequency	0.89	0.77	0.78	0.77
Sums	0.92	0.75	0.91	0.82
Longest Runs	0.89	0.78	0.77	0.78
Runs	0.92	0.76	0.89	0.82
Sp-800 Serial	0.93	0.73	0.97	0.83
Shannon	0.75	0.86	0.50	0.63
Chi-Square	0.90	0.74	0.86	0.79
Mean	0.86	0.77	0.70	0.73
Monte Carlo	0.82	0.58	0.64	0.61
Serial Byte	0.81	0.23	0.91	0.37

**Table 7 entropy-24-01503-t007:** Accuracy Results from phase two tests, lower entropy file types.

	Block Frequency	Frequency	Sums	Longest	Runs	SP-800 Serial	Shannon	chi	Mean	Monte Carlo	Serial Byte
bin	1.00	1.00	1.00	1.00	1.00	1.00	1.00	1.00	1.00	1.00	1.00
css	0.64	0.85	0.90	0.88	0.92	0.99	1.00	0.98	1.00	1.00	1.00
csv	0.93	0.98	0.99	0.99	0.99	1.00	1.00	1.00	1.00	1.00	1.00
dll	1.00	0.98	1.00	0.94	0.98	1.00	0.85	0.99	0.99	0.96	0.99
doc	1.00	1.00	1.00	1.00	1.00	1.00	1.00	1.00	1.00	0.99	1.00
elf	1.00	1.00	1.00	1.00	1.00	1.00	1.00	1.00	1.00	1.00	1.00
eps	1.00	1.00	1.00	1.00	1.00	1.00	1.00	1.00	1.00	1.00	1.00
exe	1.00	0.98	1.00	0.99	1.00	1.00	0.95	1.00	0.97	0.97	1.00
html	1.00	1.00	1.00	1.00	1.00	1.00	1.00	1.00	1.00	1.00	1.00
ics	1.00	1.00	1.00	1.00	1.00	1.00	1.00	1.00	1.00	1.00	1.00
JAVASCRIPT	0.68	0.80	0.82	0.95	0.84	0.99	1.00	1.00	1.00	1.00	1.00
json	0.72	0.71	0.90	0.95	0.94	0.99	1.00	1.00	1.00	1.00	1.00
powershell	0.93	0.97	0.98	0.99	0.98	1.00	1.00	1.00	1.00	1.00	1.00
svg	0.99	1.00	1.00	0.99	1.00	1.00	1.00	1.00	1.00	1.00	1.00
txt	0.88	0.94	0.96	0.95	0.94	0.98	1.00	1.00	1.00	1.00	1.00
xls	1.00	1.00	1.00	1.00	1.00	1.00	1.00	1.00	1.00	1.00	1.00
xml	0.31	0.44	0.65	0.98	0.48	1.00	1.00	1.00	1.00	1.00	1.00

**Table 8 entropy-24-01503-t008:** Accuracy Results from phase two tests, high entropy file types.

	Block Frequency	Frequency	Sums	Longest	Runs	SP-800 Serial	Shannon	chi	Mean	Monte Carlo	Serial Byte
7zip	1.00	0.77	0.99	0.93	0.95	1.00	0.00	0.99	0.05	0.14	1.00
APK	1.00	0.98	1.00	0.98	1.00	1.00	0.77	1.00	0.87	0.82	1.00
bmp	1.00	0.99	1.00	1.00	1.00	1.00	1.00	1.00	0.99	0.99	1.00
docx	1.00	0.98	1.00	0.99	1.00	1.00	0.87	1.00	0.95	0.91	1.00
dwg	1.00	1.00	1.00	1.00	1.00	1.00	1.00	1.00	0.99	0.99	1.00
epub	1.00	0.88	1.00	0.94	0.96	1.00	0.29	1.00	0.73	0.53	1.00
gif	1.00	0.95	0.99	0.79	0.98	1.00	0.77	1.00	0.98	0.98	0.98
gz	1.00	0.98	1.00	0.75	0.99	1.00	0.00	0.98	0.83	0.92	0.95
jpg	1.00	0.85	1.00	0.92	0.94	1.00	0.52	1.00	0.89	0.88	0.99
mkv	1.00	1.00	1.00	0.99	1.00	1.00	0.27	1.00	0.96	0.90	1.00
mp3	1.00	0.99	1.00	1.00	1.00	1.00	0.50	1.00	0.97	0.96	1.00
mp4	1.00	0.97	1.00	0.93	1.00	1.00	0.59	1.00	0.90	0.86	1.00
ods	1.00	0.91	1.00	0.99	0.98	1.00	0.47	1.00	0.84	0.74	1.00
oxps	1.00	0.95	1.00	0.79	1.00	1.00	0.24	1.00	0.77	0.85	0.97
pdf	1.00	0.95	1.00	0.87	1.00	1.00	0.67	1.00	0.92	0.89	0.99
png	1.00	0.80	0.97	0.85	0.92	0.99	0.35	1.00	0.69	0.79	0.99
ppt	1.00	0.99	1.00	1.00	1.00	1.00	0.92	1.00	0.93	0.89	1.00
pptx	1.00	0.97	1.00	1.00	1.00	1.00	0.54	1.00	0.83	0.81	1.00
RANSOMWARE-BLACKMATTER	0.74	0.75	0.73	0.78	0.75	0.71	0.91	0.62	0.77	0.54	0.15
RANSOMWARE-CONTI	0.81	0.84	0.82	0.86	0.84	0.82	0.96	0.83	0.86	0.63	0.22
RANSOMWARE-DARKSIDE	0.95	0.95	0.93	0.94	0.94	0.93	0.98	0.93	0.93	0.70	0.31
RANSOMWARE-DHARMA	0.50	0.86	0.82	0.87	0.86	0.77	0.87	0.82	0.81	0.62	0.23
RANSOMWARE-GANDCRAB	0.92	0.97	0.95	0.97	0.97	0.94	0.99	0.96	0.94	0.72	0.33
RANSOMWARE-HELLOKITTY	0.04	0.09	0.04	0.13	0.04	0.01	0.45	0.01	0.15	0.15	0.01
RANSOMWARE-MAZE	1.00	1.00	0.99	1.00	1.00	0.99	1.00	0.99	0.97	0.73	0.37
RANSOMWARE-NETWALKER	0.09	0.14	0.09	0.18	0.10	0.07	0.45	0.08	0.18	0.15	0.02
RANSOMWARE-NOTPETYA	0.76	0.77	0.75	0.79	0.77	0.75	0.84	0.76	0.78	0.59	0.22
RANSOMWARE-PHOBOS	0.73	0.87	0.85	0.87	0.87	0.85	0.87	0.86	0.84	0.62	0.24
RANSOMWARE-RYUK	0.99	1.00	0.98	1.00	1.00	0.98	1.00	0.99	0.97	0.74	0.36
RANSOMWARE-SODINOKIBI	1.00	1.00	0.99	1.00	1.00	0.99	1.00	0.99	0.97	0.74	0.36
rar	1.00	0.98	1.00	0.61	0.98	0.99	0.00	0.97	0.89	0.96	0.96
tar	1.00	1.00	1.00	1.00	1.00	1.00	0.83	1.00	0.96	0.89	1.00
tif	1.00	1.00	1.00	1.00	1.00	1.00	0.90	1.00	1.00	1.00	1.00
webp	0.88	0.28	0.67	0.21	0.56	0.78	0.15	0.63	0.40	0.55	0.93
xlsx	1.00	0.98	1.00	0.99	1.00	1.00	0.77	1.00	0.95	0.93	1.00
zip	1.00	0.75	0.99	0.95	0.97	1.00	0.00	1.00	0.08	0.20	1.00
zlib	1.00	0.98	1.00	0.75	0.99	0.99	0.00	0.98	0.82	0.92	0.95

**Table 9 entropy-24-01503-t009:** Combined test performance metrics.

	Accuracy	Recall	Precision	F1
Shannon	0.79	0.86	0.55	0.67
Chi-Square	0.91	0.74	0.86	0.79
Mean	0.89	0.77	0.76	0.76
Monte Carlo	0.84	0.58	0.71	0.64

## Data Availability

www.napierone.com.

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
