# Peer review of "Comparison of Entropy Calculation Methods for Ransomware Encrypted File Identification"

_entropy, 2022, doi:10.3390/e24101503_

Round 1
Reviewer 1 Report (New Reviewer)
The authors propose a through evaluation of different statistical methods for encrypted file identification. To carry out this evaluation, the authors use the NapierOne dataset, which seems appropriate for the task, since it includes a large variety of common file formats. The authors select a wide range of tests for their evaluation, providing a sound base for further studies and methodological comparisons in the area.
Overall, the contribution is good and useful for the field.
Some minor details about the presentation that should be fixed before publication:
- algorithm 1 seems to be redundant, given the explanation in text and Figure 1. It should be removed
- accuracy, precision, recall and F1 are well-known concepts, there is no need to provide formula definition for them.
- Figure 3 seems a bit confusing. Why have a single color (red) for < 80%? If you want to use this block-based format to present the results, why not use color shading to represent different values? see for instance how confusion matrices are shaded
- For citations [22], [77], authors should update their references to the most recent version of the publications:
[22] De Gaspari, F., Hitaj, D., Pagnotta, G. et al. Reliable detection of compressed and encrypted data. Neural Comput & Applic (2022). https://doi.org/10.1007/s00521-022-07586-7
[77] De Gaspari, F., Hitaj, D., Pagnotta, G. et al. Evading behavioral classifiers: a comprehensive analysis on evading ransomware detection techniques. Neural Comput & Applic 34, 12077–12096 (2022). https://doi.org/10.1007/s00521-022-07096-6
Author Response
Please see attached document for our responses

Reviewer 2 Report (New Reviewer)
I appreciate the intention to create a survey-like extensive work to assess the usability of almost all entropy measurement methods for detecting decrypted content.
The authors perform a multi-step evaluation of these measurement techniques and then provide a shortlist of techniques that are performing reasonably well on the considered testing dataset.
The idea and work are not particularly novel. The work by Casino et al. [1] has done a preliminary evaluation of the NIST suite and afterward picked up the best performing tests from the suite in order to build a compression vs. encryption detector.
This work expands on that methodology to evaluate a larger amount of entropy measurement tests. Nevertheless, it is a well-motivated work and can serve as a general point of reference for the use of statistical tests in the tasks of encryption detection.
The paper is easy to read, and the results are presented fairly well. Some of the pictures are unreasonably large (maybe due to the format), and they create a bit of a gap in following up the text.
Minor comments:
The resolution of some figures should be improved. Particularly Figure 4 is very soft and can get tiring to look at, plus it negatively impacts the overall presentation given that some images are of adequate resolution. Nevertheless is sth that can be easily fixed.
Overall I am leaning towards accepting this work.
References
[1] Casino, Fran, Kim-Kwang Raymond Choo and Constantinos Patsakis. “HEDGE: Efficient Traffic Classification of Encrypted and Compressed Packets.” IEEE Transactions on Information Forensics and Security 14 (2019): 2916-2926.
Author Response
Please see attached document

This manuscript is a resubmission of an earlier submission. The following is a list of the peer review reports and author responses from that submission.
Round 1
Reviewer 1 Report
This paper did an experiment over various measures of randomness on a large dataset to test which one is good for diferetiating encrypted data (for ransomware detection purpose).
My main problem with this paper is the research contribution. Apart from the fact that such an experiment on such dataset has not been done before and some conclusions, e.g., high accuracy for setial byte correltation, the suggestion of combination of measures, there is no/little contribution to scientific research study.
The entrire paper is describing the experiemnt settings, w/ some explanations over existing measures/results/techniques. I don't see, e.g., any new measures proposed that better than previous, or a new ransomware detection technique.
I think the authors need to highlight what is their contribution to information theory.
The presentation can also be improved, e.g., section 4&5 can be merged, i.e., present the results right after decription of each phase.
Reviewer 2 Report
The work assesses which entropy-measure strategy--- among the many that exist theoretical or implemented in toolkits and libraries--- is capable of distinguishing files that have been "encrypted" by ransomware from others that as well have high entropy (e.g., compressed files).
Among the merit of the work, there is the comprehensive list of strategies considered and the selection of a known, large, and stable data set, NapierOne.
The experimental design is fair, with standard measures such as FP, FN, TP, TN, accuracy etc.; so are the experiments for the most part, except the last part where metrics are combined, trying to find a combination that works better than the strategies considered separately; this part sounds a bit unplanned.
That said, there is a "but", which makes this reviewer sceptical about the contribution of the work; here, I do not mean "contribution" in terms of data and experiments, which are considerable, but rather in terms of the relevance of the project and its outcome.
The paper builds its importance on the assumption that files "encrypted" by ransomware have high entropy as files that are sheerly encrypted. This hypothesis seems weakly posed. The paper itself reports from the state of the art that ransomware tries to tamper with the files they output to lower entropy just to escape detection. One of the papers cited [7] even brings evidence that entropy-measuring strategies fail to detect the file produced by ransomware.
And this negative conclusion is also what the work also finds out, but this conclusion was not conceived to repeat [7]'s experiment, with better insights and more details. Instead, it looks simply unsurprising due to a weak, not well-posed, hypothesis.
Also, the focus on ransomware makes the experimental setting chosen look weak. The data set, NapierOne, which is comprehensive of several types of files with low and high entropy, does contain files that have been encrypted by a few ransomware, but this is a very limited selection of ransomware families.
Since the focus is on ransomware, one expected much more data, e.g., files coming from a much larger set of ransomware-type.
We leave the authors with two suggestions to improve the paper:
- try to discuss more extensively the limitation of entropy detection strategies per se and position your paper better (e.g., by accepting a different hypothesis that ransomware's output can be low entropy).
- extend the data set with more families of cryptographic ransomware.
Reviewer 3 Report
Reviewer Comments to Authors
This article should be rejected. The authors have proposed a novel method to attempt to compare the file entropy values generated by crypto-ransomware more appropriately. They have spent lots of time producing this paper, which definitely has values in the fields of mathematics and statistics. It could otherwise be a very good piece of work in mathematics and statistics. However, in my opinion, this paper should be rejected, because the authors made too many claims that were fundamentally incorrect from the point of view of cybersecurity. The manuscript can only demonstrate that the authors' chosen algorithms worked best on their chosen dataset, which was in fact developed by themselves in their earlier work. The authors did not seem to have understood the latest ransomware trends and its attack mechanisms properly. Given that the first author, S.R.Davis is a PhD student in cybersecurity, I recommend any future submissions should ideally go to cybersecurity-themed journals to face scrutiny from reviewers with more cybersecurity-related backgrounds.
MAJOR ISSUES:
(1) "Ransomware" and "crypto-ransomware" are not interchangeable. Ransomware is not just crypto-ransomware executable files trying to encrypt user files (data loss). Some also start to perform information exfiltration (data leak). File encryption is now an optional feature of ransomware. In 2021, the top 3 ransomware groups, Conti, Avaddon and REvil, accounted for 60% of attacks and all performed information exfiltration. Information exfiltration does not alter file entropy values.
(2) This study lacks internal validity, because their methods contained several flaws, and their results won't apply to other ransomware attack vectors. In line 44, authors claimed ransomware would leave characteristic information in its encrypted files. However, the Memento ransomware is now using WinRAR to perform password-protected file encryption with compression. WinRAR is very likely to be considered benignware by authors. In reality, there is no way to differentiate entropy changes from WinRAR encryption initiated by the users vs by ransomware. In figure 1 on page 7, the authors decided there was a threshold to distinguish encrypted vs non-encrypted file. Where is the logical evidence for that? What about WinRAR-initiated encrytion and compression 2-in-1? In their refererence [7], it was stated that ransomware could perform partial encryption to minimize entropy changes; as littile as 10% could be encrypted to make almost no difference, but this was not discussed in the submitted manuscript. Since [7], BlackMatter, DarkSide and LockBit use partial encryption. In their refererence [7], it was also mentioned that ransowmare could perform base64-encoding of encrypted file contents to "normalize" the entropy values; again it was not discussed in their submitted manuscript.
(3) This study lacks external validity, because using only their self-constructed dataset like NapierOne seemed like a self-fulfilled prophecy. Anyone can run any algorithms on any dataset, and obtain results like accuracy and false positive rates. However, there is no guarantee that the files attacked by crypto-ransomware collected by the authors in their own NapierOne dataset were a fair representation of ransomware artifacts in the wild, not to say that it is sometimes impossible to obtain ransomware artifacts of human-operated ransomware, or ransowmare performing information exfiltration. The quality of their dataset is questionable; all 4 ransomare variants were very only and were all crypto-ransomware. If authors wish to perform a pure mathematical evaluation, they should at least apply their methods to another 3rd-party dataset to cross-validate their results.
(4) The authors did not acknowledge that file types were merely containers, and that the file contents did not have to match file extension names. Check the concept of "file type tunneling"
(5) It felt like authors prioritised file entropy changes at the main method for ransomware mitigation. Although authors did not explicitly declared that their paper could contribute to ransomware mitigation, they way they constructed their paper tend to made readers feel that way. If it is the case, ransowmare mitigation is a complicated matter and requires a multi-layered approach. The feature of file encryption is already outdated.
(6) This manuscript uses too many industry sources as references, many of which have not been peer-reviewed. It is possible that some of them have been written out of commercial interest, thus less reliable.
(7) This study does not have self-disclosed limitations or suggestions for future research. Quoting from the second recommended reading: “The limitation of a study is the systematic bias that was not or could not be controlled by researchers and could affect the study results inappropriately”. “Because no research is perfect, existing research should consider including an adequate discussion of its limitations in its submitted manuscript and suggest future research directions. Such an adequate coverage of research limitations should include discussions of both internal validity (accurately measures what it intends to measure) and external validity (the sample results accurately represent the results of the entire target samples)”
OTHER ISSUES:
(1) In line 32, "a recurring theme". This is outdated information
(2) In line 36-38, entropy values will be reduced if base-64-encoded or partially encrypted
(3) In line 132 - 134. How does this contribute to the knowledge of ransomware and its mitigation?
RECOMMENDED READING FOR AUTHORS:
(1) This survey paper to properly understand the latest trends in ransomware evolution.
McIntosh, T., Kayes, A. S. M., Chen, Y. P. P., Ng, A., & Watters, P. (2021). Ransomware Mitigation in the Modern Era: A Comprehensive Review, Research Challenges, and Future Directions. ACM Computing Surveys (CSUR), 54(9), 1-36. DOI https://doi.org/10.1145/3479393
It has summarized some common inadequacies of ransomware-related research, some of which unfortunately apply to this manuscript.
(2) This paper describes ransomware from a cyber kill chain model.
Dargahi, T., Dehghantanha, A., Bahrami, P. N., Conti, M., Bianchi, G., & Benedetto, L. (2019). A Cyber-Kill-Chain based taxonomy of crypto-ransomware features. Journal of Computer Virology and Hacking Techniques, 15(4), 277-305. DOI https://doi.org/10.1007/s11416-019-00338-7
(3) Human-Operated Ransomware by Microsoft
https://docs.microsoft.com/en-us/security/compass/human-operated-ransomware
Round 2
Reviewer 3 Report
The authors cannot and should not claim that their research will improve the anti-ransomware landscape, or this paper should be rejected because it contains fundamental errors from the perspective of cybersecurity. I suggest that the authors should resubmit this to a cybersecurity-themed journal if they are so confident that their method would work.
In the authors' response to the first review, they claimed "The authors envision that this research will aid crypto-ransomware detection technique developers who intend to use entropy as an indication of infection, which entropy calculation methods are best suited for their application and also provide a word of caution that entropy on its own is not an ideal solution when testing against a large and varied representation of commonly used file types. "
The counter-arguments are:
(1) Many antivirus software products already have folder protection against ransomware encryption. Check Microsoft "Controlled Folder Access".
(2) What's the point of detecting encryption, if the authors cannot differentiate user-initiated encryption from ransomware-initiated encryption?
(3) The fact that the authors admitted they didn't know the Memento ransowmare was disappointing, as it was one of the first which used benign encryption software to masquerade their activities.
(4) The fact that the authors admitted they didn't know "file type tunnelling" in ransowmare was disappointing. It was mentioned in several landmark ransomware studies, including:
Continella, A., Guagnelli, A., Zingaro, G., De Pasquale, G., Barenghi, A., Zanero, S., & Maggi, F. (2016, December). Shieldfs: a self-healing, ransomware-aware filesystem. In Proceedings of the 32nd annual conference on computer security applications (pp. 336-347).
Kharraz, A., Robertson, W., Balzarotti, D., Bilge, L., & Kirda, E. (2015, July). Cutting the gordian knot: A look under the hood of ransomware attacks. In International conference on detection of intrusions and malware, and vulnerability assessment (pp. 3-24). Springer, Cham.
Kharraz, A., & Kirda, E. (2017, September). Redemption: Real-time protection against ransomware at end-hosts. In International Symposium on Research in Attacks, Intrusions, and Defenses (pp. 98-119). Springer, Cham.
(5) Even applying entropy-based detection itself is not new. See the 3 studies aforementioned in (4).
(6) Crypto-ransomware is so old. 3 years ago, they dominated the landscape. Now most ransomware samples started to focus on information exfiltration. The world has moved on from crypto-ransomware! Investivating something that is being deprecated is disappointing. Academic research should aim to create new knowledge, not reinventing the wheels.
(7) The following 2 surveys are recommended to the authors. Both published in ACM Computing Surveys (impact factor >14)
Oz, H., Aris, A., Levi, A., & Uluagac, A. S. (2021). A survey on ransomware: Evolution, taxonomy, and defense solutions. ACM Computing Surveys (CSUR).
McIntosh, T., Kayes, A. S. M., Chen, Y. P. P., Ng, A., & Watters, P. (2021). Ransomware mitigation in the modern era: A comprehensive review, research challenges, and future directions. ACM Computing Surveys (CSUR), 54(9), 1-36.